# Evaluation of Machine Learning Interatomic Potentials for the Properties of Gold Nanoparticles

**DOI:** 10.3390/nano12213891

**Published:** 2022-11-03

**Authors:** Marco Fronzi, Roger D. Amos, Rika Kobayashi, Naoki Matsumura, Kenta Watanabe, Rafael K. Morizawa

**Affiliations:** 1University of Technology Sydney, Ultimo, NSW 2007, Australia; 2Australian National University, Canberra, ACT 2601, Australia; 3Fujitsu Limited, Kawasaki 211-8588, Japan

**Keywords:** machine learning potentials, gold clusters, molecular dynamics, structures, heat capacities

## Abstract

We have investigated Machine Learning Interatomic Potentials in application to the properties of gold nanoparticles through the DeePMD package, using data generated with the *ab-initio* VASP program. Benchmarking was carried out on Au${}_{20}$ nanoclusters against *ab-initio* molecular dynamics simulations and show we can achieve similar accuracy with the machine learned potential at far reduced cost using LAMMPS. We have been able to reproduce structures and heat capacities of several isomeric forms. Comparison of our workflow with similar ML-IP studies is discussed and has identified areas for future improvement.

## 1. Introduction

There is currently a lot of interest in Machine Learning Interatomic Potentials (ML-IP) as a method showing promise of approaching the high accuracy of *ab-initio* methods, while remaining closer to the cost of empirical classical approaches [1,2]. Classical molecular dynamics (MD) approaches, even though very fast, are not very accurate, and commonly limited in that the force fields cannot break bonds and thus cannot study reactions. Although some work have been done in order to build potentials that can describe chemical reactions, the results are not yet satisfactory [3,4]. Furthermore, many interesting properties, require very long MD runs—much longer than are currently practical with density functional theory (DFT) methods.

ML-IP have been in existence for several years [5,6] but have gained increased popularity in recent years from the availability of systematic workflows through the development of various software packages such as DeepMD [7,8], Medea [9] or lammps-polymlp [10]. Recent successes in this direction, include application to problems in phase-change materials for memory devices [11], nanoparticle for catalysts [12], carbon-based electrodes for chemical sensing [13] and electrolyte solutions design [14]. This has inspired us to explore ML-IP methodology, starting with a system we are familiar with, to evaluate the feasibility of tackling larger problems such as reactions on surfaces and catalysis.

Close colleagues have been interested in gold nanoparticles for many years. There are a variety of questions regarding their structure and properties. Ford et al. [15] have previously studied small clusters using DFT, and more recently larger clusters have been studied with classical MD methods, see e.g., Ref. [16], amongst others. There is scope to investigate a wider range of structures, for larger systems, and including properties such as the temperature dependence, which require much longer time scales than can be achieved with standard DFT methods. The catalytic properties of gold clusters have been known for some time—a recent volume of Chemical Reviews [17] was devoted to this, and contains several relevant articles pointing out that the smaller clusters are actually more active catalysts [18]. An advantage of developing ML-IP potentials is that they can be additive i.e., it is not necessary to repeat the work for the gold potential, instead additional terms describing the molecule being studied can be added.

There are now a few works in the literature looking at ML potentials for gold clusters it e.g., Refs. [19,20,21,22,23]. However, they all differ in a variety of ways, such as investigating different properties or using different ML-IP programs and models or slightly different methodology so part of this study is to compare our approach with these others to evaluate the best approach. Furthermore, as this methodology is still relatively new there are still many unknowns, such as the number of points or underlying methods required for training. To this end we have thus carried out a series of exploratory investigations to gain an understanding of the methodology for use in future more complex studies.

## 2. Method

The ML-IP package chosen for this study was DeePMD [7,8] developed by Wang and co-workers to build potential energy models through deep learning. It is a Python/C++- based package interfaced to the TensorFlow deep learning framework [24]. The potential is developed using the local environment of each atom described in terms of distances and angles to all other atoms within a given radius. This chemical environment is described mathematically as a descriptor [25], and definition of the descriptor is something that distinguishes the various ML-IP packages. These descriptors are then fitted to provided data through some machine learning mechanism, which in the case of DeePMD is a neural network. Details of the neural network are given in their paper [7]. In short, DeePMD leverages the standard tensor operations provided by TensorFlow through a feedforward network using atomic positions, energies and forces as input. The neural network structure consists of 5 hidden layers and is trained by the Adam stochastic gradient descent method to output energy and forces. An example of the parameter file used for training is provided in the Appendix A.

To generate the ML-IP it is necessary to produce the forces for a variety of small clusters in a selection of configurations. A total of a few hundred representative configurations, which means several thousand individual forces, should be adequate. We started from non-equilibrium structures of Au${}_{20}$. The reason for having non-equilibrium structures is that the eventual aim is to study the melting of gold clusters, and this will require that the potential describes regions that may be well away from the most stable geometries. Accordingly, we used the structures of Au${}_{21}$, referred to as isomers I, III and IV, given in Ref. [26] but removing one atom, and we refer at them as structure I, II and III. This gave the structures shown in Figure 1 (coordinates provided in Appendix A). One of these closely resembles the tetrahedral global minimum of Au${}_{20}$, but with distortions from the purely tetrahedral geometry. The other two structures have a cage-like form and a more open flatter form, respectively. Note that related studies have mostly looked at larger clusters [20,21,22,23], primarily at structures only, but as we wanted to benchmark against an *ab-initio* MD run we chose this cluster as being computationally manageable but still able to cover the properties of interest.

Molecular Dynamics calculations were then performed on each structure to generate a series of geometries and their associated forces to parameterise the force field. We used Density Functional Theory (DFT) calculations within the framework of the Vienna Ab Initio Simulation Package (VASP) [27,28]. The projector augmented-wave (PAW) potentials are used to describe the core–valence interaction, with the valence electrons described by periodic plane waves with cut-off energy of 520 eV. The cubic supercell with lattice parameter of 30 Å contains the 20-atom clusters of the three structures. We used the generalized gradient approximation (GGA) for the exchange–correlation functional as formulated by Perdew-Burke-Ernzerhof (PBE) [29]. The energy and forces tolerance for structural relaxation are 10${}^{-6}$ eV and 10${}^{-2}$ eVÅ${}^{-1}$ respectively. Equations of motion were integrated using the velocity Verlet algorithm, with a time step of 2.5 fs. The temperature of the system was stabilized using a Nosé–Hoover thermostat with the temperature-damping parameter of 10 time steps. The Nosé-Hoover thermostat is a deterministic method (no random forces or velocities) that is based on the extended system idea, in which an additional degree of freedom s is introduced to represent the thermal reservoir, which ensure an ergodicity within the canonical ensemble [30].

The data in the required format (energies, forces, geometries) is extracted from the VASP OUTCAR file using the DeePMD module dpdata [31]. This data is randomly split into training and validation sets in the ratio of 80:20. The DeePMD training program dp is run on the training set using the descriptor se_e2_a and parameters based on those provided in the DeePMD examples [31]. This process can be run on all the structures individually or all combined.

## 3. Results and Discussion

### 3.1. Generating the Data

Using VASP molecular dynamics simulations, we collected data for a maximum of ∼8500 timesteps for structure I and to ∼7500 for structure II and III, corresponding to 21,250 fs, and 18,750 fs respectively. The quality of the simulation is indicated by the excellent conservation of the constant of motion and the negligible drift in the fictitious kinetic energy of the electrons for simulations of that duration (Figure 2). The average temperature is ∼800 K for each structure, and the relatively large deviation due to the coupling with the Nosé thermostat, ensure an extensive sampling over the potential energy surface at large atomic distances (see Table 1). The large temperature fluctuations, indicated in Figure 2 and Table 1 correspond to a large nuclei’s velocities which lead to a large fluctuations in atomic distances during the simulations, ensuring an extensive sampling over the potential energy surface at large atomic distances. Table 1 reports the statistical analysis of the temperature distribution over time, indicating the amount of simulation-time where the system is away from the average temperature.

A low average energy and small deviation during the simulation suggest the larger stability of structure II compared to the other configurations (see Table 2).

### 3.2. Evaluating the Potential

The training procedure in DeepMD reports RMS errors on both the force and the energy for the training and validation sets. The errors reported were similar for both training and validation, meaning that any overfitting had been avoided. The resulting potential from the training was tested on the validation set. Not all structures could be represented equally well. For example, for structure I, using the potential trained only on the energies and forces resulting from calculations on structure I, taking 1000 random points from the validation set produces the following results:
#1000 points randomly chosen from the validation set#DEEPMD INFO  # number of test data: 1000#DEEPMD INFO
Energy RMSE: 8.219273e-02 eV#DEEPMD INFO
Energy RMSE/Natoms: 4.109637e-03 eV#DEEPMD INFO
Force RMSE: 6.739765e-02 eV/A#DEEPMD INFO
Virial RMSE: 3.285809e+00 eV#DEEPMD INFO
Virial RMSE/Natoms: 1.642904e-01 eV
and plotted in Figure 3.

It shows quite a good fit with RMSE of 0.082 eV and a coefficient of determination, R${}^{2}$, of 0.960. The correlation plots for structures II and III also given in the Figure. These figures were produced using potentials obtained by training only on data from structure II and structure III respectively. Note that the fit for structure II is the best of the three. This may be because structure II closely resembles the pyramidal form of Au${}_{20}$ which is the global minimum and more rigid than the others.

It is possible to use all the data simultaneously and train to produce a more general potential which can describe all three structures. The fit for all structures trained simultaneously is shown in Figure 4a. The fit remains excellent with an R${}^{2}$ coefficient of 0.975, which is not quite as good as structure II on its own but bear in mind this covers a larger conformation space. A scatter plot (Figure 4b) of the errors in the energy shows that the vast majority of the errors are between ±0.1 eV but there are a few outliers deviating by as much as 0.4 eV.

In molecular dynamics the forces are as important as the energy. Most machine learning studies in the area of potential fitting tend to discuss the energy more than the forces but as we believe the accuracy of the forces is meaningful we also include these graphs (Figure 4c,d). Figure 4c gives the ratio of the magnitude of the predicted force to the VASP-calculated force. In an ideal world this would be unity. Looking at the graph it is reassuringly very strongly peaked at 1. Furthermore, the magnitude of the angle between the computed and the predicted force, which should be zero, is similarly strongly peaked at zero. There are a small number of forces which have an error in angle greater than 100 degrees but these are too rare to show up in the histogram.

The most basic test for a potential is what it predicts as the minimum energy structure. To do this we compared the structures as predicted by a geometry optimisation in VASP with the 0K equilibrium structures as predicted by LAMMPS [32] using the machine learned potentials (Figure 5). The VASP minimum energy structures, starting from the initial structure described above, are in Figure 6.

Structure II is the global minimum structure. The coordinates of these final structures are also given in the Appendix A. Relative to the pyramidal structure, structure III is 0.744 eV higher, and structure I 1.845 eV higher.

For the structures produced by LAMMPS, structure II again optimises to the global minimum, and structure III gives a geometry the same as VASP. Isomer I, from the same starting geometry as used with VASP, optimises to a structure very similar to VASP’s local minimum, but with a smaller energy difference relative to the ground state. Structure III is now 0.671 eV above the global minimum, which is in reasonable agreement with the VASP result. These results were all obtained with a potential derived using data from all structures simultaneously, i.e the one whose statistics are shown in Figure 4. Cao et al. [23] studied structures of several small gold clusters including Au${}_{20}$, and found a local minimum which resembles the one shown here. Ford et al. [15] also found a similar structure at approximately the same energy above the global minimum.

### 3.3. Using the Potential

We computed the vibrational density of states from the velocity auto-correlation function in the three structures (see Figure 7). Each spectrum shows a single pronounced peak at ∼1–1.5 THz, reproducing one of the main peaks measured experimentally. This is due to the small size of the nanoparticle considered here, that favours low frequency phonon modes due to the large surface-bulk ratio, as suggested by Bertoldi et al. [33]. However, here we focus on the accuracy of DeePMD models in reproducing ab-initio calculations, and the absence of high frequency phonon modes is a negligible detail.

Using the energy fluctuation of the system with respect to the temperature described by Labastie et al., we calculated the heat capacity as follows [34]:
(1)$$C_{v}=\frac{<\delta E^{2}>-<\delta E{>}^{2}}{k_{B}T^{2}}\;,$$
where *E* is the energy of the system, $k_{B}$ is the Boltzmann constant and *T* is the average temperature.

Using equation Equation (Equation 1), the specific heat capacity has been calculated using time windows of 2000 steps starting from different points of MD simulations. For structure II, $C_{v}$ value oscillates between 33.89 J/K/mol, or 0.34 meV/K/Atom, and 36.66 J/K/mol, or 0.38 meV/Atom (around 10%), with an average of 35.12 J/K/mol or 0.36 meV/K/Atom. Similar results are obtained for structure I and III with a relatively larger fluctuation in the values (around 20%) and the results are summarized in Table 3.

The specific heat values are higher than the corresponding bulk and decrease with the size of the nanoparticle [35]. Also, a second high peak frequency becomes less pronounced for small cluster size [36]. The literature report values of C${}_{p}∼$ 30, when calculated by modified tight-binding second-moment approximation [37], and ∼16.6 J/K/mol when calculated by Density Functional based Tight Binding method [38]. Small fluctuations in the $C_{v}$ value during the *ab-initio* simulation timeline, confirm the ergodicity of the systems, and the small deviation from the experimental value indicate the accuracy of the simulations.

When C${}_{v}$ is calculated by frozen phonon models in (1 × 1 × 1) expansion of the supercell. Due to the low symmetry of the clusters, the 126 single point calculations needed to evaluate the $C_{v}$ require CPU Time: 01:00:57. However, the value obtained is one order of magnitude larger (∼450–500 J/K/mol), due to the presence of intense fictitious low frequency phonon modes. The large error may be partially recovered using a larger expansion of the supercell, which, however, will lead to a very heavy calculation due to the low symmetry configuration of the clusters. The phonon DOS is included in the Appendix A.

Using the DeePMD calculated interatomic potentials, we repeat the nanocluster molecular dynamics with the same setup used for VASP MD. We use the interatomic potentials extrapolated by using all the three structures as training/test sets, which naturally include a generalised description of the of the energy surface. The deviation from average values of temperature and energy are comparable to the ones calculated in VASP MD for the same structure within 0.25% for the temperature and 6% for the energy, respectively (see Figure 8 and Table 4). The calculated $C_{v}$ value using energy fluctuations is around 40% smaller that the ones calculated using VASP. However, this deviation is comparable with the distribution of values using different theoretical/computational approaches reported in the literature [37,38]. Furthermore, the vibrational DOS calculated using ML-IP, reproduce the low frequency peak calculated by VASP MD (see Figure 9).

The computational time necessary for a simulation to reach 8500 timesteps using VASP is (CPU Time: 10,301:11:16) on 48 CPUs, whereas the same number of timesteps uses only (CPU Time: 00:03:20) on 4 CPUs using the Machine Learning Potential in LAMMPS, confirming the huge resource-wise benefit of ML approach if compared to conventional quantum mechanical models, preserving accuracy.

## 4. Conclusions

Using the DeePMD program we have shown that Machine Learned potentials are a feasible way of calculating the properties of gold nanoclusters at similar accuracy but reduced computational cost to standard *ab-initio* molecular dynamics methods. Our potential correctly reproduces the global minimum of Au${}_{20}$ and one of the local minima. We have calculated the heat capacity using both *ab-initio* and machine-learned MD and they agree satisfactorily with each other and related studies from the literature. In the course of the investigation further scope for improvement became apparent, such as reducing the number of data points while retaining accuracy, using alternative methods to generate the training data, better descriptors for the ML model and so on. These are currently under investigation. In particular, recent works [20,39] using new techniques that build on and enhance existing data to produce more general data, such as in active or delta learning, look promising for future work.

## Figures and Tables

**Figure 1 nanomaterials-12-03891-f001:**

Structures of Au${}_{20}$ used as starting geometries for generating data to train the machine learning potential.

**Figure 2 nanomaterials-12-03891-f002:**
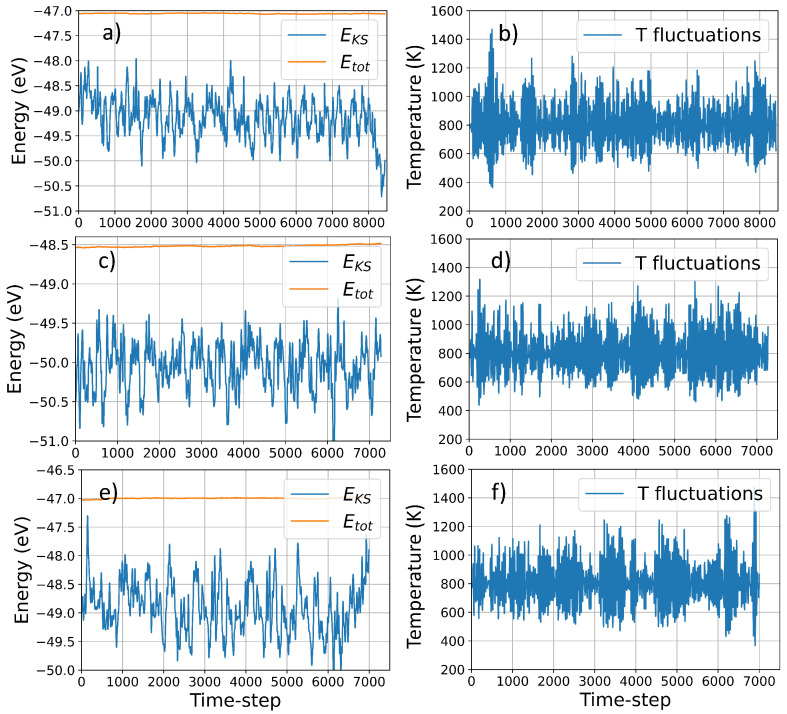
Temperature and energy (potential vs total) fluctuations during VASP MD simulation structure I, II and III, In the figure, E${}_{KS}$ and E${}_{tot}$ indicate the electronic energy and total energy, respectively, whereas T fluctuations indicate the temperature fluctuations during the simulations due to the Nosé thermostat coupling. Panel (**a**) and (**b**), (**c**) and (**d**), and (**e**) and (**f**) correspond to structure I, II and III, respectively.

**Figure 3 nanomaterials-12-03891-f003:**
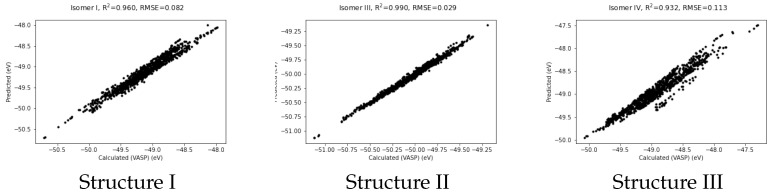
Predicted vs VASP-calculated energies (in eV) for structures I, II and III.

**Figure 4 nanomaterials-12-03891-f004:**
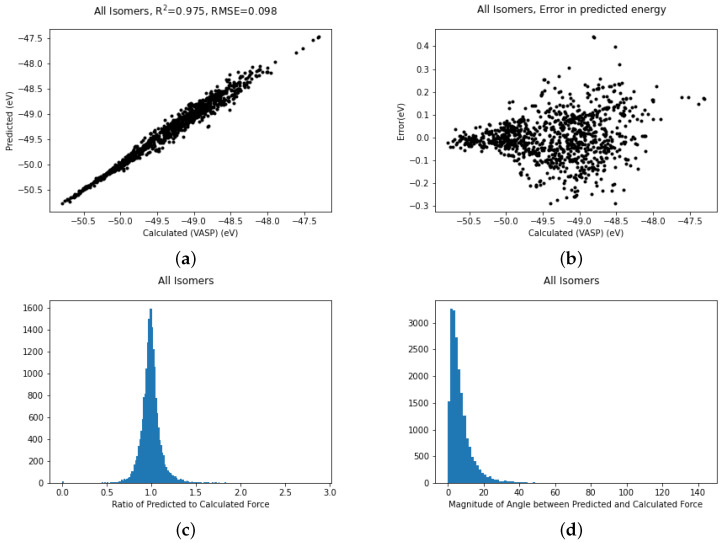
Plots of (**a**) Fits of energies, (**b**) scatter in energy error, (**c**) error in magnitude of forces and (**d**) error in direction of forces for all the gold cluster structures trained simultaneously.

**Figure 5 nanomaterials-12-03891-f005:**

LAMMPS optimised structures using the machine learned potential. Energies (in eV) given relative to the lowest energy structure.

**Figure 6 nanomaterials-12-03891-f006:**

VASP optimised structures from a regular VASP geometry optimisation. Energies (in eV) given relative to the lowest energy structure.

**Figure 7 nanomaterials-12-03891-f007:**
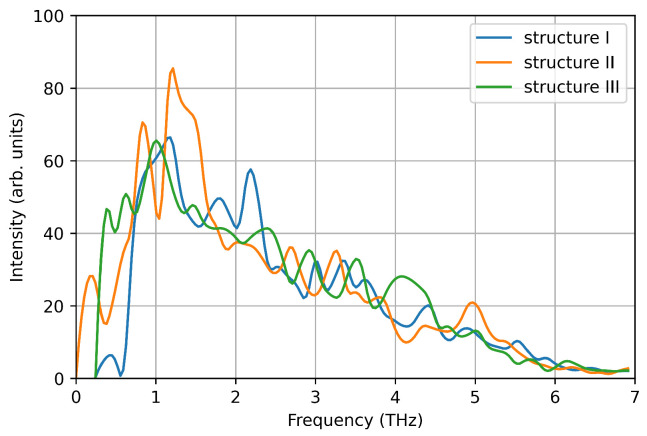
Vibrational density of states computed from the velocity auto-correlation function in VASP simulations.

**Figure 8 nanomaterials-12-03891-f008:**
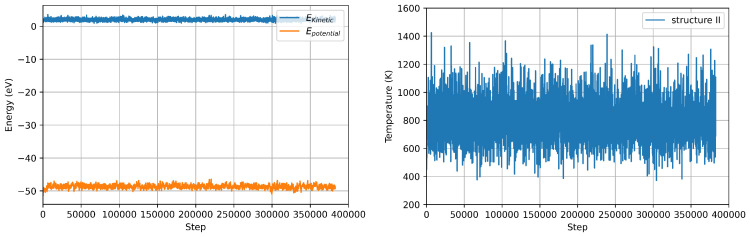
Energy (potential vs total) (left panel) and Temperature (right panel) fluctuations during LAMMPS MD simulation structure II.

**Figure 9 nanomaterials-12-03891-f009:**
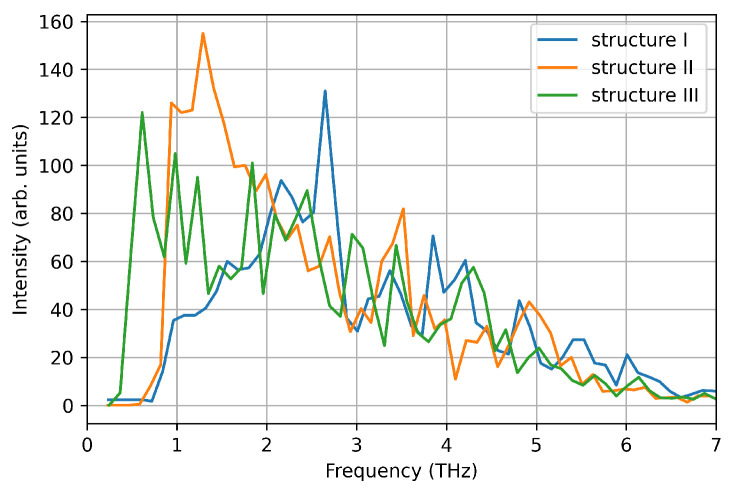
Vibrational density of states computed from the velocity auto-correlation function in LAMMPS simulations.

**Table 1 nanomaterials-12-03891-t001:** Statistical analysis of thermal fluctuations in VASP MD simulations.

	Structure I	Structure II	Structure III
mean	799.50	799.73	799.42
std	143.91	146.47	144.87
min	362.59	438.78	367.98
25%	696.80	691.54	699.67
50%	792.57	790.09	793.40
75%	890.36	898.34	888.90
max	1470.42	1318.70	1471.67

**Table 2 nanomaterials-12-03891-t002:** Statistical analysis of energy fluctuations in VASP MD simulations.

	Structure I	Structure II	Structure III
mean	−49.13	−50.05	−48.91
std	0.40	0.30	0.44
min	−50.71	−51.10	−50.25
25%	−49.39	−50.35	−49.23
50%	−49.12	−50.98	−48.94
75%	−48.85	−49.81	−48.63
max	−47.95	−49.25	−47.34

**Table 3 nanomaterials-12-03891-t003:** Specific heat capacity (C${}_{v}$) at 800 K calculated using equation Equation (Equation 1) for VASP and LAMMPS simulations (values are given in J/K/ mol).

Structure	(C${}_{v}$) (VASP)	(C${}_{v}$) (LAMMPS)
I	29.12	19.92
II	35.12	20.63
III	39.12	24.44

**Table 4 nanomaterials-12-03891-t004:** Statistical analysis of energy fluctuations in structure II during LAMMPS MD simulations.

	Temperature (K)	Total Energy (eV)
mean	797.08	−46.70
std	152.98	0.67
min	371.95	−49.78
25%	688.94	−47.16
50%	786.95	−46.72
75%	894.24	−46.25
max	1424.30	−43.92

## Data Availability

Coordinates of the structures used in this study are provided in the Appendix A.

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
