# Peer review of "Evaluation of Machine Learning Interatomic Potentials for the Properties of Gold Nanoparticles"

_nanomaterials, 2022, doi:10.3390/nano12213891_

Round 1

Reviewer 1 Report

The authors present an investigation where by using DeePMD program they have shown that Machine Learned potentials are a feasible way of calculating the properties of gold nanoclusters, essentially, Au20, at commonly achieved accuracy but  reducing computational cost to standard ab initio molecular dynamics methods.

The paper in well presented and well organized. However, the authors confused the term nanoparticles with nanoclusters. They are managing nanoclusters, so I suggest to resolve this possible source of confusion.

Another crucial question, how scalable is their method compared to larger clusters? or up to which clusters can their method work? detailing the answer.

Other remarks: teh abstract should clarify in much more detail what has been done and done by the authors.

The term "gold nanoparticles" shoudl be substituted by the acronym AuNPs in the text.

After such mandatory changes the paper could be suitable for a publication.

Author Response

Dear Editor,
Thank you very much for considering the paper, and the reviewers for their comments. We have addressed them as follows.

Reviewer 1:

The paper is well presented and well organized. However, the authors confused the term nanoparticles with nanoclusters. They are managing nanoclusters, so I suggest to resolve this possible source of confusion.
Thank you to the referee for pointing this out. We hadn’t realised the difference but I see now that it is very important to state we are working with nanoclusters because of the size of our system. We will keep the title as nanoparticles as that is the ultimate aim of this project but we are benchmarking our calculations via nanoclusters. We have also kept the term nanoparticles when referring to related works which are investigating nanoparticles.  We have made the appropriate changes throughout the text.

Another crucial question, how scalable is their method compared to larger clusters? or up to which clusters can their method work? detailing the answer.

This is a very good question and one we are currently trying to address. Because we were trying to gain an understanding of the method, we did thorough benchmarking of a reasonably-sized computationally manageable nanocluster. How extensible it is, is certainly a question we are interested in. The amount of calculation necessary to test the scalability of the model is quite large. It is beyond the scope of the current study as project continuation is contingent on evidence of progress through publication, and we are leaving this work for a following study.

Other remarks: the abstract should clarify in much more detail what has been done and done by the authors.

We have added more detail to the abstract.
“We have investigated Machine Learning Interatomic Potentials in application to the properties of gold nanoparticles through the DeePMD package, using data generated with the ab initio VASP program. Benchmarking was carried out on Au20 nanoclusters against ab initio  molecular dynamics simulations and show we can achieve similar accuracy with the machine-learned potential at far reduced cost using LAMMPS. We have been able to reproduce structures and heat capacities of several isomeric forms. Comparison of our workflow with similar ML-IP studies is discussed and has identified areas for future improvement.”

The term "gold nanoparticles" should be substituted by the acronym AuNPs in the text.

After we corrected the text as in point 1 to differentiate between gold nanoparticles and nanoclusters, the term “gold nanoparticle” ended up only appearing once in text, so we have left it in expanded form.

Reviewer 2 Report

The manuscript entitled “Evaluation of machine learning interatomic potentials for the properties of gold nanoparticles” by Marco Fronzi et al. presented a study that utilized DeePMD to calculate the properties of gold nanoclusters. 

The proposed method reproduced the global minimum of Au20 nanoclusters reasonably. The authors also calculated and compared the heat capacity using both ab initio and machine-learning approaches. However, I have some questions that need to be addressed before publication. 

I would recommend a major revision of the manuscript for the authors to answer the following questions:

(1) On line 53, the authors mentioned: “… which in the case of DeePMD is a neural network”. I would suggest the authors to provide more descriptions of the DeePMD neural network. For example, what's the neural network structure? What are the neural network input and output? What's the training process in this study? 

(2) On line 99, the sentence “… ensure an extensive sampling over the potential energy surface at large atomic distances (see Tab.1).” is not clear. Could authors provide more explanations on the interpretation of Tab.1.’s data?

(3) Since Figure 3 and Figure 4 present the same information for three structures, why did the authors display the two figures separately? I would suggest the authors to combine these two figures together. 

(4) For Figure 5b, what’s the definition of error here? Could authors provide some explanations or equations? Also, the caption “Energy scatter” seems misleading. Please consider changing it to terms such as “energy prediction error”?

(5) For line 142 to line 144, what’s the potential reason that LAMMPS optimizes to a global minimum while VASP only optimizes to a local minimum? I suggest authors to provide more discussions and explanations on this observation. 

I suggest the authors to proofread the manuscript again for any typos and minor issues. The following typos and minor issues should be addressed before publication:

a. On line 32, the “Chemical Reviews (volume 120, issue 2, 2020)” should be cited as a reference. 

b. Please remove the question mark in the citations for DeepMD [7?] throughout the manuscript. 

c. On line 86, please correct the question mark in the citation for “dpdata [?]”. 

d. On line 89, please correct the question mark in the citation for “DeePMD examples [?]”. 

e. For Figure 2, I suggest the authors to re-arrange the subplots. Readers cannot see x labels clearly with the current display (x labels should be time step). I also suggestion authors to clearly mark each subplot with (a) to (e), then describe the meaning for each subplot in figure caption.

f. On line 119, the sentence “… coefficient of 0.98,” should be changed to 0.975 to be consistence with the number formats within this manuscript. 

Author Response

Reviewer 2:

(1) On line 53, the authors mentioned: “… which in the case of DeePMD is a neural network”. I would suggest the authors to provide more descriptions of the DeePMD neural network. For example, what's the neural network structure? What are the neural network input and output? What's the training process in this study? 

We have now provided a more detailed description of the neural network process.
“ Details of the neural network are given in their paper [7]. In short, DeePMD leverages the standard tensor operations provided by TensorFlow through a feedforward network using atomic positions, energies and forces as input. The neural network structure consists of 5 hidden layers and is trained by the Adam stochastic gradient descent method to output energy and forces. An example of the parameter file used for training is provided in the supplementary information”

(2) On line 99, the sentence “… ensure an extensive sampling over the potential energy surface at large atomic distances (see Tab.1).” is not clear. Could authors provide more explanations on the interpretation of Tab.1.’s data?

“The large temperature fluctuations, indicated in Fig.2 and Tab.1, correspond to a large nuclei’s velocities which lead to large fluctuations in atomic distances during the simulations, ensuring an extensive sampling over the potential energy surface at large atomic distances.  Tab.1 reports the statistical analysis of the temperature distribution over time, indicating the amount of simulation time where the system is away from the average temperature.”

(3) Since Figure 3 and Figure 4 present the same information for three structures, why did the authors display the two figures separately? I would suggest the authors to combine these two figures together. 

Originally, we had intended to discuss the figures separately and in more detail but looking on it now we agree it does look silly.  We have now combined them into one figure.

(4) For Figure 5b, what’s the definition of error here? Could authors provide some explanations or equations? Also, the caption “Energy scatter” seems misleading. Please consider changing it to terms such as “energy prediction error”?

The definition of error is the difference between calculated and predicted values. We think this is a standardly accepted definition. We will change Energy scatter to “Scatter in the energy error”

(5) For line 142 to line 144, what’s the potential reason that LAMMPS optimizes to a global minimum while VASP only optimizes to a local minimum? I suggest authors to provide more discussions and explanations on this observation. 
Yes, this was something that puzzled us greatly such that we followed it up later and discovered we had a mistake in our input. We had repeated the calculation for reproducibility but when we further examined the individual steps of the optimisation we realised that the starting structure was not the intended one - not Structure II (which it optimised to) but not the Structure I as was used for the LAMMPS optimisation either. We have now repeated all calculations and Structure I now optimises to the expected structure. This does not change the overall findings of our paper but at least does resolve the curiosity which was interesting but not significant to our conclusions.

The text now reads:
“Isomer I, from the same starting geometry as used with VASP, optimises to a structure very similar to VASP’s local minimum, but with a smaller energy difference relative to the ground state” We have also updated Figure 6.

I suggest the authors to proofread the manuscript again for any typos and minor issues. The following typos and minor issues should be addressed before publication:

  1. On line 32, the “Chemical Reviews (volume 120, issue 2, 2020)” should be cited as a reference. 

As this was a general reference to a whole issue rather than an individual paper we weren’t sure how to reference it. We were following: https://blog.apastyle.org/apastyle/2012/09/citing-a-whole-periodical.htm
where it recommends citing in-text. We have found an alternative format and have now changed it to

Astruc, D. (Ed.). (2020). Nanoparticles in Catalysis  [Special issue]. Chemical Reviews, 120(2).

  1. Please remove the question mark in the citations for DeepMD [7?] throughout the manuscript. 
  2. On line 86, please correct the question mark in the citation for “dpdata [?]”. 
  3. On line 89, please correct the question mark in the citation for “DeePMD examples [?]”. 

Apologies for this sloppiness. This has now been fixed.

  1. For Figure 2, I suggest the authors to re-arrange the subplots. Readers cannot see x labels clearly with the current display (x labels should be time step). I also suggestion authors to clearly mark each subplot with (a) to (e), then describe the meaning for each subplot in figure caption.

This has now been done.

  1. On line 119, the sentence “… coefficient of 0.98,” should be changed to 0.975 to be consistence with the number formats within this manuscript.
    This has now been done.

We believe we have now addressed all the reviewers’ concerns and hope our manuscript is now acceptable.

Round 2

Reviewer 1 Report

In the revised version, the authors addresed all the questions raised during the first reviewing activity. I suggest that the paper now can be published.

Reviewer 2 Report

For comment (1)(2)(3)(4), the authors have addressed my comments. For comment (5), the authors claimed they have corrected the mistake in the input, and the results look reasonable after revision. The authors have corrected all the typos and minor issues as well. 

Therefore, I believe the revised manuscript has appropriately addressed all my questions. I would recommend its publication on Nanomaterials.